# Development of a Zeolite H-ZSM-5-Based D-μSPE Method for the Determination of Organophosphorus Pesticides in Tea Beverages

Bing Bai [1,†], Nan Wu [1,†], Haifeng Yang [1,2], Haiyan Liu [1,2], Xiaofen Jin [1,2], Lei Chen [1,2], Zhiying Huang [1,2], Changyan Zhou [1], Shouying Wang [1,2,*] and Wenshuai Si [1,2,3,*]

1   Institute for Agri-Food Standards and Testing Technology, Shanghai Academy of Agricultural Sciences, 1000 Jinqi Road, Shanghai 201403, China
2   Shanghai Co-Elite Agri-Food Testing Technical Service Co., Ltd., 1000 Jinqi Road, Shanghai 201403, China
3   School of Health Science and Engineering, University of Shanghai for Science and Technology, 516 Jungong Road, Shanghai 200093, China
*   Correspondence: wangsy@saas.sh.cn (S.W.); siwenshuai@saas.sh.cn (W.S.)
†   These authors contributed equally to this work.

**Abstract:** In this study, a novel dispersive micro-solid phase extraction (D-μSPE) technique with H-ZSM-5 zeolite as an adsorbent was developed for the determination of 21 trace pesticides in tea beverages. The adsorption and desorption of H-ZSM-5 zeolites were investigated based on structural characteristics and adsorption properties similar to those of H-beta zeolites. In combination with the properties of the adsorbates, it was explained that the adsorption reaction occurred on the microporous surface and mesopores of H-ZSM-5. Based on optimal parameters, the beverage samples were extracted by 50 mg of zeolite within 1 min. The zeolite was eluted with 2 mL of an acetonitrile-water mixture after separation, and the eluent was filtered prior to HPLC-MS/MS analysis. The D-μSPE protocol demonstrated acceptable accuracy and precision, with recoveries between 62.1% and 106.6% and relative standard deviations of 1.4% to 12.6%, as validated by analytical reliability. The correlation coefficient in the linear range of 0.2–50 ng·mL$^{-1}$ was greater than 0.98, with limits of detection of 0.05–0.1 ng·mL$^{-1}$ and limits of quantification of 0.1–0.2 ng·mL$^{-1}$. The matrix effects ranged from 76.2% to 112.7%. The results indicate that the novel D-μSPE technique based on H-ZSM-5 is a rapid, simple, green and economical method for the determination of pesticide residues in tea beverages. The proposed method achieved simultaneously low adsorbent dosage, 20-fold enrichment factor, rapid pre-concentration in 12 min, minimal organic wastes, and effective reduction of matrix interference.

**Keywords:** zeolite H-ZSM-5; tea beverage; organophosphorus pesticides; dispersive solid-phase extraction; HPLC-MS/MS

## 1. Introduction

Tea consumption is second only to water, the most popular beverage in the global market [1]. As an easily absorbed aromatic beverage, it offers a wealth of health benefits and is an ideal remedy for neurological disorders and cardiovascular disease [2,3]. However, problems such as excessive pesticide residues, microbial contamination, and the overuse of food additives still exist. Previous studies have reported that some organophosphorus (OPs) are frequently detected on tea in some of the major tea-producing countries, although about 70% of them are banned due to their high toxicity [4–7]. Hence, the management of pesticides in tea and its products is necessary and ongoing to reduce chronic dietary exposure. The detection of pesticide residues in tea usually involves extraction and cleanup. Tea beverages are subject to interference from pigments, alkaloids and polyphenols [8]. The dilution of tea beverages also increases the difficulty of determining harmful residues

in processed products. Compared to traditional QuEChERS procedures, the dispersive solid phase extraction (D-SPE) technique, where the adsorbent is dispersed into the sample solution, allows for extraction and enrichment of the target [9,10]. D-SPE does not require much consumption of organic waste, and adsorption and desorption can be achieved more quickly due to the increased exposure of the adsorbent to the solvent [11,12]. However, cheap, stable and reusable adsorbents with high selectivity remain a major problem for the D-SPE technique.

The adsorbent material of interest in this study is synthetic zeolite. The main mechanisms of zeolite H-Beta are the electrostatic interaction between the adsorbent and the functional groups of pesticide molecules based on previous research [13]. The preparation process requires the replacement of sodium ions in zeolite with ammonium ions, followed by roasting to obtain H-type zeolite, so both H-Beta and H-ZSM-5 have the characteristics of acidic zeolites with high porosity [14–16]. Regarding the topologically specifically, Beta contains two 12-membered ring 3D linear channel systems with pore openings of $7.7 \times 6.6$ Å and a third channel system of $5.6 \times 5.5$ Å, while ZSM-5 is composed of eight 5-membered ring structural units mirror connected to form a corrugated mesh layer with 10-membered rings that cross each other with pore sizes of $5.6 \times 5.3$ Å and $5.5 \times 5.1$ Å, respectively [17]. H-Beta showed good analytical performance in the adsorption of 35 pesticides in water, but the main focus of the study was on some new low-toxicity pesticides [13]. Zeolite H-ZSM-5, which is simple to prepare, structurally stable, non-polluting, and low-cost, develops curved pore channels in its internal structure, facilitating the separation of organic macromolecules in complex matrices. Therefore, H-ZSM-5 is a potential adsorbent for the adsorption of common highly toxic pesticides (e.g., organophosphorus) in tea.

The aim of this study was to establish a novel technology for the detection of trace pesticides in complex tea beverages. A highly efficient D-µSPE method was explored and validated for the determination of 16 OPs and five other pesticides in tea beverages using zeolite H-ZSM-5 as the adsorbent. A series of analytical parameters, including the amount of zeolite, adsorption time, desorption solvent and desorption time were optimized to achieve sensitive and accurate determination of 21 pesticides in tea beverages under optimal D-µSPE procedure, and the reusability of zeolite H-ZSM-5 was studied. Although our proposed D-µSPE with multi-target simultaneous analysis cannot cover all organophosphorus pesticides, this study not only provides a promising solution for the analysis of pesticides in complex beverages and the rapid enrichment of trace chemicals but also places higher demands on the future development of novel dispersive extraction techniques as green analytical techniques.

## 2. Materials and Methods

### 2.1. Reagents and Materials

ZSM-5 was prepared with the molar composition of 1 $SiO_2$: 0.025 $Al_2O_3$: 0.06 $Na_2O$: 0.10 Butylamine (BA): 10.0 $H_2O$. Silica gel powder as a silica source for zeolite synthesis was supplied by Qingdao Guichuang Fine Chemical Co., Ltd. (Qingdao, China). Sodium meta-aluminate ($NaAlO_2$) used as the aluminum source was provided by Sinopharm Chemical Reagent Co., Ltd. (Shanghai, China), composed of 39.5% $Na_2O$ and 48.0% $Al_2O_3$. The structure-directing agent is tetraethylammonium hydroxide (TEAOH, 25%) (Kente Catalysts Inc. Hangzhou, China). Sodium hydroxide (NaOH, Sinopharm Chemical Reagent Co., Ltd., Shanghai, China) was used as the base source. Ultrapure water with a resistance of 18.2 MΩ.cm (25 °C) was prepared by Milli-Q Preference (Molsheim, France). All standard solutions with certificates were purchased from Alta Scientific Co., Ltd. (Tianjin, China) at a concentration of 100 mg·L$^{-1}$ in acetonitrile, stored at $-18$ °C according to the certificate. The intermediate stock standard of 5 ug·mL$^{-1}$ was diluted with methanol and stored at $-18$ °C with six months of expiry. Chromatographic grade methanol and acetonitrile used for HPLC-MS/MS analysis were obtained from Merck (Darmstadt, Germany).

### 2.2. Preparation of Zeolite H-ZSM-5 Adsorbent

Zeolite H-ZSM-5 was prepared with minor modifications based on a previous report [18]. Generally, 0.836 g of NaOH, 2.125 g of NaAlO$_2$ and 2.926 g of BA were dissolved with 72 g of H$_2$O. Then, 24 g of silica gel powder was added with stirring and stirred for 2 h. The mixture was heated in a Teflon-lined stainless-steel autoclave at 170 °C for two days. The autoclave was cooled to room temperature after the formation of crystals. The obtained solid product was separated by filtration and washed with deionized water. The deionized water and structure-directing agent were then removed by overnight drying at 100 °C and calcination at 550 °C for 6 h, respectively. Finally, H-ZSM-5 was successfully prepared by ion exchange of the calcined zeolite with 1M NH$_4$Cl solution (1:50 g·mL$^{-1}$) at 85 °C for 2 h and by calcination at 550 °C for 6 h.

### 2.3. Characterization of Zeolite H-ZSM

The parameters for characterization were determined in accordance with previous reports on novel adsorbents [19,20]. The Rigaku Ultima IV X-ray diffractometer (XRD) was used to confirm the phase purity of the synthesized H-ZSM-5 with nickel-filtered Cu K$\alpha$ radiation ($\lambda$ = 0.15418 nm). Hitachi S-4800 SEM was operated at an accelerating voltage of 3 kV to obtain two SEM images (3 kV × 10 k, 3 kV × 50 k). The nitrogen adsorption-desorption isotherm at 196 °C was measured on a BELSORP-max device. The prepared zeolite H-ZSM-5 was vented under vacuum at 300 °C for 6 h prior to measurement. The values ranging from 0.01 to 0.15 were employed to calculate the specific surface area (S$_{BET}$) of zeolite H-ZSM-5 based on the Brunauer, Emmett and Teller (BET) principle. The total pore volume (V$_t$) was estimated by calculating the nitrogen adsorption at p/p0 of 0.99. The zeolite was dissolved in HF solution and quantified by Thermo IRIS Intrepid II XSP inductively coupled plasma-atomic emission spectrometry (ICP-AES) to determine the molar ratio of SiO$_2$/Al$_2$O$_3$ for H-ZSM-5.

### 2.4. HPLC-MS/MS Analysis

The chromatographic parameters referred to the previous study with slight modifications by comparing the intensity and separation of peak to obtain the best chromatographic separation and rapid analysis [20], with an ACQUITY BEH-C18 column (2.1 × 100 mm, id:1.7 μm, Waters, Milford, IN, USA) and a Waters H-Class UPLC system (Waters, Milford, IN, USA) in tandem with a QqQ mass spectrometer (AB4500, AB SCIEX, Framingham, MA, USA) to collect retention and MRM profiles for 21 pesticides. The electrospray ionization source (ESI) parameters were consistent with previous studies. Water (A) containing 0.1% formic acid (FA) and 5 mmol·L$^{-1}$ ammonium acetate and methanol (B) were used as mobile phases. Samples were eluted at a flow rate of 0.38 mL·min$^{-1}$ as follows: phase A was equilibrated at 5% before 1.2 min; the volume fraction of phase A was increased from 5% to 95% from 1.2 to 4.5 min and maintained at this concentration until 6.0 min; the volume fraction of phase A was reduced from 95% to 5% from 6.0 to 6.8 min; 5% A was maintained from 6.8 min to the next analysis. The chromatographic separation of one sample was completed over 8 min with an injection volume of 3 μL. The column temperature was 45 °C. The MRM parameters for the 21 pesticides were used to identify and quantify the target analytes in Table S1.

### 2.5. D-μSPE Process

The target analytes were adsorbed using a dispersed solid-phase material and then eluted with a suitable solvent so that the analytes were enriched and cleaned in one go [13]. Firstly, 50 mg of zeolite H-ZSM-5 was put into a 50 mL centrifuge tube with 40 mL tea beverage, which was vortexed at 2500 rpm for 1 min to adsorb the target analytes as much as possible. Subsequently, the zeolite H-ZSM-5 was separated from the aqueous phase by centrifugation at 3200 g·min$^{-1}$ for 3 min, and then desorbed by vortex for 3 min and ultrasonication for 1 min using 2 mL of acetonitrile-water in equal volume. The eluent

was separated from the adsorbent by centrifugation at 3200 g·min$^{-1}$ for 3 min and then transferred to a vial through a 0.22 μm membrane filter for HPLC-MS/MS analysis.

### 2.6. Sample Collection

The tea beverages were purchased from local stores in Shanghai randomly. Tea samples were filtered through a glass microfiber filter and kept at 4 °C within one week before the D-μSPE procedure.

### 2.7. Validation of D-μSPE

This study investigated the matrix effects, linearity, limit of detection (LOD), limit of quantification (LOQ), accuracy, and precision of the optimal D-μSPE method on a case-by-case basis with reference to the updated guidance document SANTE/11312/2021 [21]. The limits of detection (LODs) were determined by spiking experiments when more than 83% of the samples (at least five out of six spiked replicate samples) were detected with acceptable ionic ratio deviations and intensities greater than 3-fold signal-to-noise (S/N). The LOQ was set as double the LOD to ensure the accuracy while its intensity was $\geq 10 \times$ S/N [22]. The linearity range was studied at seven concentrations of 0.2, 0.5, 1.0, 5.0, 10.0, 25, and 50 ng·mL$^{-1}$. The accuracy and precision of the method were validated at the lowest LOQ, $5 \times$ LOQ, and $25 \times$ LOQ levels. The matrix effect of D-μSPE was calculated with a percentage ratio of standard in the tea beverage extract to the counterparts in the solvent [20,23].

### 2.8. Regeneration of Zeolite H-ZSM-5

A washing procedure was executed before the reuse to remove the residues of zeolite H-ZSM-5 from the prior sample preparation. Specifically, 5 mL of acetonitrile was used to wash zeolite H-ZSM-5 with a vortex at 2000 rpm/min for 5 min, and sonication for 5 min. The process was repeated twice after centrifugation at 3200 g·min$^{-1}$ for 5 min. The last supernatant was filtered through a 0.22 μm nylon membrane and analyzed by high-performance liquid chromatography-tandem mass spectrometry. The H-ZSM-5 was washed thoroughly when the concentration of the pesticide was less than the method limits of quantification in the supernatant. After washing, the zeolite H-ZSM-5 was air-dried in a ventilator overnight and used for subsequent reuse.

## 3. Results and Discussion

### 3.1. Characterization of Prepared Zeolite H-ZSM-5

The results of XRD, N$_2$ adsorption-desorption, and SEM are shown in Figure 1. Compared with the diffraction peaks reported on the MFI topology [14], the XRD pattern of H-ZSM-5 showed comparable high-resolution diffraction peaks at ~8°, 9°, and 23° without any impure crystalline phases (Figure 1A). The N$_2$ adsorption-desorption characteristics further confirmed the satisfactory preparation of the zeolite H-ZSM-5 with the marked type IV isotherm (Figure 1B). Specifically, steep N$_2$ uptake was observed in the relative pressure range of p/p0 < 0.1, corresponding to the filling of N$_2$ in the micropores. The adsorption volume was moderate with a mild hysteresis loop in the range p/p0 > 0.1, which was characteristic of mesopores. The high surface area and S$_{BET}$ indicated a large number of micropores in H-ZSM-5 in the small crystal size of H-ZSM-5. The micropore surface area (S$_{mic}$) and volume (V$_{mic}$) were 330.8 m$^2$/g and 0.15 cm$^3$/g, respectively. The properties, 364.1 m$^2$/g of S$_{BET}$, 33.3 m$^2$/g of external surface area (S$_{ext}$), 0.25 cm$^3$/g of V$_t$ and 0.10 cm$^3$/g of mesoporous volume (V$_{meso}$), are slightly lower than those of the reported H-Beta zeolite, whereas indicated the porosity and sieving properties of H-ZSM-5 similar to H-Beta zeolite (Table 1). As observed in the SEM images, the prepared zeolite H-ZSM-5 was composed of homogeneous ellipsoidal crystals with a particle size of ~1 μm, which aggregated to form primary particles with a diameter of approximately 100 nm (Figure 1C,D). The SiO$_2$/Al$_2$O$_3$ molar ratio of the prepared H-ZSM-5 was ~34.6 determined by ICP-AES. The agreeable properties of H-ZSM-5 suggested its high potential as an adsorbent.

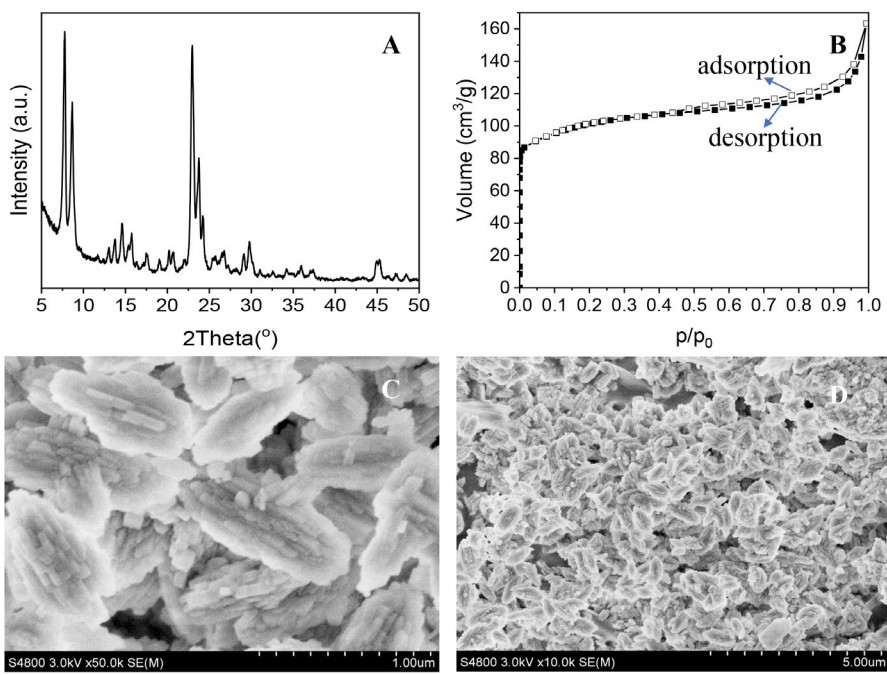

**Figure 1.** XRD pattern (**A**), N$_2$ adsorption–desorption isotherm (**B**) and SEM images with different magnifications (**C**,**D**) of the prepared zeolite H-ZSM-5.

**Table 1.** The properties of the prepared zeolite H-ZSM-5 and H-Beta.

| Title 1 | H-ZSM-5 | H-Beta [13,20] |
|---|---|---|
| $S_{BET}$ (m$^2$/g) | 364.1 | 600 |
| $S_{mic}$ (m$^2$/g) | 330.8 | 530.6 |
| $V_{mic}$ (cm$^3$/g) | 0.15 | 0.21 |
| $V_t$ (cm$^3$/g) | 0.25 | 0.36 |
| $V_{meso}$ (cm$^3$/g) | 0.10 | 0.15 |
| $S_{ext}$ (m$^2$/g) | 33.3 | 69.4 |
| $SiO_2$/$Al_2O_3$ | ~34.6 | ~36.0 |

### 3.2. Sample Volume

The adequate solvent volume provides H-ZSM-5 more opportunities to interact with absorbates. The large sample volume could widen the enrichment factor, while too large reduces the adsorption rate due to the target diffusing over a larger spatial area. The H-ZSM-5 is a nanomaterial with a greater density than water and tends to settle without external force conditions. The performance of 5 mL, 10 mL, 20 mL and 40 mL were studied under the equal adsorbent dosage. The adsorption rates of five pesticides, including bupirimate, butachlor, coumaphos, disulfoton sulfoxide and profenofos, were less than 60% at 5 mL of water, while the percentages were significantly improved when the volume increased to 40 mL (Figure 2A). More than ten analytes (e.g., demeton, disulfoton and phorate sulfoxide) maintained good adsorption and desorption at any sample volume, with recoveries ranging from 80% to 97%. Only the rate of disulfoton sulfoxide showed a decrease by 14% from a volume of 20 mL to 40 mL. The higher volume would decrease the efficiency of multiple simultaneous sample processing. Hence, 40 mL of sample was used in the preparation. In Cao's study [12], a sample volume of 5 mL was taken for the adsorption of neonicotinoid insecticides in environmental water using UiO-66. Although some SPE techniques have been reported to allow higher sample volumes [24], they are usually limited by the inability to pass the SPE cartridge too quickly which reduces the concentration efficiency.

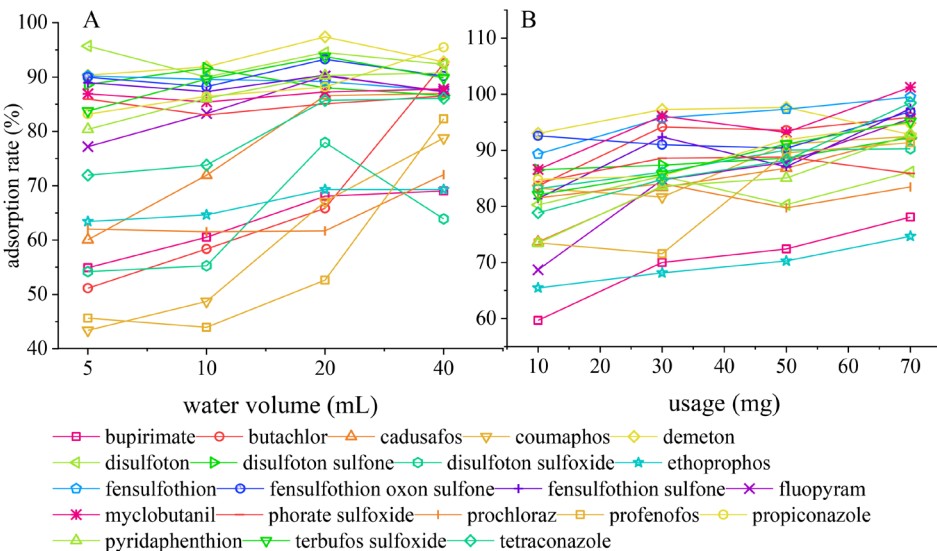

**Figure 2.** The effect of sample volume (**A**) and absorbent dosage (**B**).

### 3.3. The Amount of Zeolite H-ZSM-5

In addition to affecting the efficiency of adsorption, the amount of adsorbent could indicate the performance of the zeolite H-ZSM-5, which cannot be ignored in the evaluation of new adsorbents. Four H-ZSM-5 usages were tested in 40 mL of water. As shown in Figure 2B, a dosage of 10 mg was slightly inadequate for the complete adsorption of 21 pesticides. The adsorption rates of bupirimate, ethoprophos, and fluopyram were below 70%. At the usage of 50 mg, all targets were effectively recovered with values ranging from 70% to 100%, while the overall adsorption rates did not change significantly when further increasing the zeolite dosage. The adsorption percentages of studied pesticides were distributed between 60% and 111%. Furthermore, 92% of these values were between 70% and 110%, showing satisfactory performance of accuracy. It showed that the dosage advantage of zeolite H-ZSM-5 is prominent among the many new materials with multiple target adsorption. The study proposed by Arnnok [25] applied modified zeolite NaY for the detection of 20 pesticides in environmental samples, 150 mg zeolite NaY was utilized to adsorb the targets in 125 mL of the sample volume with recoveries between 64 and 128%. Zeolite H-ZSM-5 could achieve the same efficiency with a minimum of 10 mg in this study. To ensure the more stable method performance, 50 mg of Zeolite H-ZSM-5 was used to pre-concentrate the tea beverage.

### 3.4. Adsorption Time and Characteristics

Mechanical shaking at 2500 rpm/min was evaluated from 1 min to 9 min to accelerate adsorption. It showed that all pesticides have excellent stable adsorption efficiencies, with a span of less than 16% between the minimum and maximum adsorption percentage at different adsorption times (Figure S1). Unexpectedly, the adsorption ratio of twenty-one targets reached a steady state in just 1 min, indicating that zeolite H-ZSM-5 is a powerful and promising adsorbent for the adsorption of a wide range of pesticides. Hence, 1 min of the vortex was chosen in the D-μSPE process. The optimum extraction time for OPs in tea samples using porous polypropylene membrane bags is 60 min in the method proposed by Sajid [26]. The magnetic SPE technique based on amino-modified multi-walled carbon nanotubes (m-MWCNTs-NH$_2$) required at least 2 min to adsorb the three pesticides completely studied [27].

Investigating the hydrophilic and lipophilic coefficients of the studied pesticides, log K$_{ow}$ (collected from the PubChem website, Table S1) with values ranging from 1.5 to 4.7. It was observed that there was no clear linearity between the log K$_{ow}$ of these compounds and the adsorption efficiency of zeolite H-ZSM-5. This finding is similar to our previous study of

H-beta zeolite. Pesticide molecule size distributes between $3.4 \times 10^{-4}$ and $5.6 \times 10^{-4}$ nm$^3$, calculated by the predicted molar volumes (PMV, Table S1) of pesticide molecules obtained from ChemSpider. Zeolite H-ZSM-5 has a 10-membered ring mesopore diameter of Ca 0.56 nm [28]. Thus, the possible sites of adsorption are inside the mesopore and on the zeolite surface.

### 3.5. The Solvent and Time of Desorption

Acetonitrile has shown better efficiency of eluting the pesticides from the zeolite in previous studies [13,20]. Therefore, four volumes (1 mL, 2 mL, 4 mL, 6 mL) were discussed for all compounds studied using acetonitrile as the eluent. In Figure 3A, 21 pesticides were recovered between 61 and 106% at all studied volumes. Two (2) mL of acetonitrile (CH$_3$CN) had better method stability as evidenced by distribution characteristics, with recoveries of 71–96%. Elution of 4–6 mL brought higher chance errors, resulting in lower recoveries of several targets. Hence, 2 mL was used to optimize further with four solvent ratios of CH$_3$CN-H$_2$O. The standards were prepared with the same ratio of solvent for quantification to avoid the difference in peaks patterns and sensitivity of the compounds diluted with different ratios of CH$_3$CN-H$_2$O. The results showed that a decrease in the acetonitrile ratio did not reduce the elution efficiency (Figure 3B). The recoveries of seventeen analytes ranged from 90 to 110% when eluting with a 1:1 mixture of CH$_3$CN-H$_2$O, except disulfoton obtained a recovery of 66%. The ratios of 7:3 and 1:1 both yielded recoveries in the range of 80–110% for nineteen pesticides. Eventually, the 2 mL of 1:1 CH$_3$CN-H$_2$O mixture was chosen as the desorbing solvent considering the environmental toxicity of the organic reagents. Our proposed D-μSPE procedure showed a significant improvement in the waste of organic solvents compared to similar previous reports. Solvent with 6 mL of ethyl acetate and 0.2 mL of methanol was used in Ma's Magnetic SPE method [29]. The DSPE procedure based on modified zeolite NaY also used a minimum of 3 mL of organic solvent mixture [25].

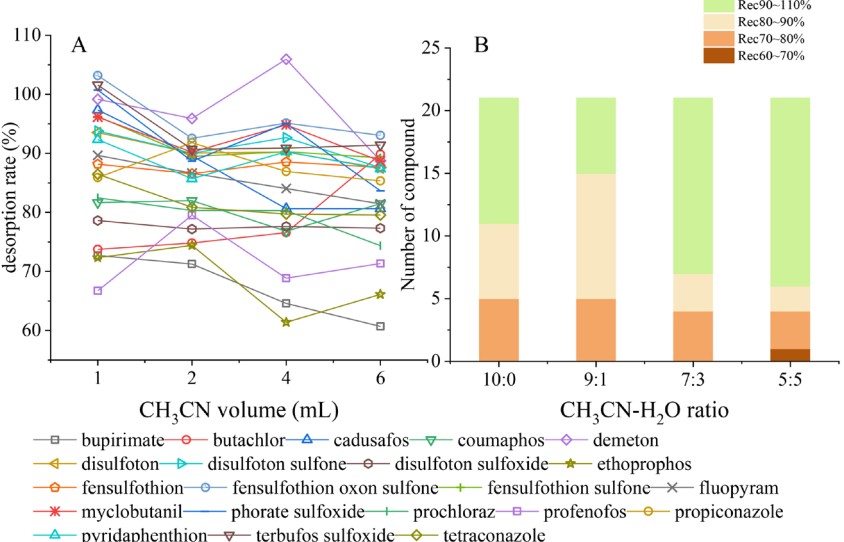

**Figure 3.** The desorption performance of CH$_3$CN volume (**A**) and mixtures of CH$_3$CN/H$_2$O (**B**).

The operating times of ultrasound and vortex were investigated to improve the efficiency of the D-μSPE procedure. The results showed that 11 pesticides, such as phorate sulfoxide, demonstrated desirable recoveries with minor fluctuations between 80% and 110% for all parameters studied, without presented in Figure 4. In Figure 4A, the ultrasonic treatment contributed to improving desorption efficiency when the time was within 3 min, but recovery of prochloraz was extremely unsatisfactory over the 5 min of ultrasound. Optimization of vortex-assisted desorption was tested at the 3 min ultrasound. The results in Figure 4B showed a slight improvement in overall recoveries with the 10 min vortex,

but similar outcomes could be expected with the 1 min vortex. Therefore, 1 min vortex and 3 min sonication were the optimized factors for desorption. It suggested that solvent transformation readily enabled the reversibility of the adsorption of zeolite H-ZSM-5.

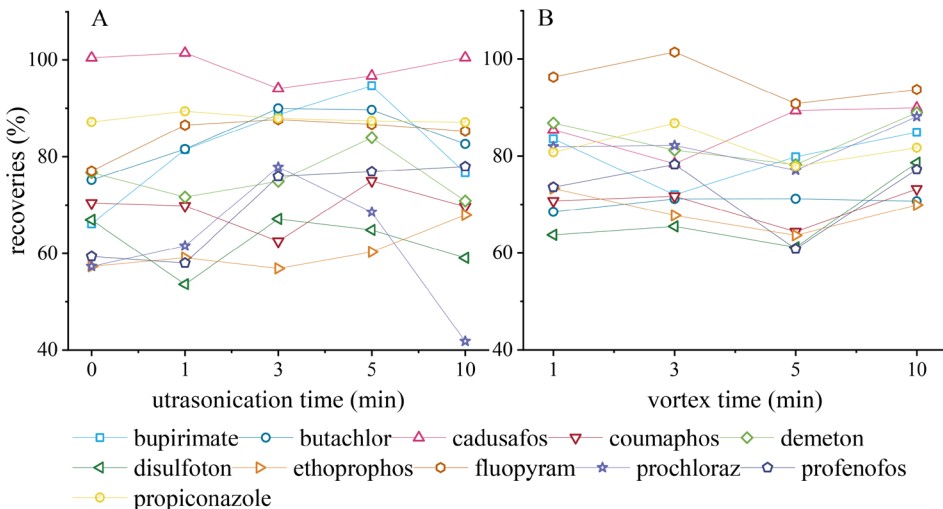

**Figure 4.** The desorption efficiency of ultrasonication (US) time (**A**) and vortex (VOR) time (**B**).

### 3.6. The Validation of D-μSPE Process

The results showed that in both black tea beverages (BT) and green tea beverages (GT), the ME of all studied pesticides distributed between 80% and 120%, except bupirimate with 76.2% and 77.6% in BT and GT, respectively (Figure S2). According to the guidelines, signal enhancement and inhibition beyond 80–120% requires correction treatment. Thus, the standard curve in solvent was acceptable for quantitation for this study. The regression curves and correlation coefficients ($R^2$) were fitted and calculated with seven concentrations. The linear coefficients were above 0.99 for 21 pesticides in the concentration range of 0.2–50 ng·mL$^{-1}$, while the $R^2$ of cadusafos was 0.98691. Table S2 provides the linear regression equation for quantitative calculations. The limit of detection (LOD) of prochloraz was 0.1 ng·mL$^{-1}$. The remaining pesticides had a LOD of 0.05 ng·mL$^{-1}$. The minimum spiked concentration (LOQ) was 0.2 ng·mL$^{-1}$ for prochloraz and 0.1 ng·mL$^{-1}$ for the remaining pesticides. Wu [30] proposed a multi-residue method for detecting pesticides in Oolong tea using QuEChERS, the LODs and LOQs of 89 pesticides were 1 to 25 μg·L$^{-1}$ and 10 to 50 μg·L$^{-1}$, respectively. A solvent-based extraction of 13 OPs in tea samples using a porous polypropylene membrane bag showed acceptable LODs in the range of 1.4–7.2 ng·g$^{-1}$ [26]. Our study indicates that satisfactory sensitivity for the determination of trace pesticides was presented in the proposed D-μSPE protocol.

The accuracy and precision of the D-μSPE method were verified at 0.1 ng·mL$^{-1}$, 0.5 ng·mL$^{-1}$, and 2.5 ng·mL$^{-1}$ in the black tea matrix (Table 2), the chromatogram of quantitation ion at 0.5 ng·mL$^{-1}$ was shown in Figure 5. Six replicates at each level were tested to calculate the relative standard deviation (RSD). The results showed that the average recoveries of 21 pesticides ranged from 62.1 to 106.6% at 0.1 ng·mL$^{-1}$ with RSDs of 2.2% to 10.8%. The spiking level of 0.5 ng·mL$^{-1}$ obtained mean recoveries of 62.3–96.3% and the RSDs between 1.4 and 9.1%. The mean recoveries were between 63.6% and 90.0%, and the RSDs were in the range of 2.1–12.6% at 2.5 ng·mL$^{-1}$. The criterion of recovery beyond the margin of 70–120% is also acceptable in a consistent deviation (RSD < 20%) according to SANTE/11312/2021. All the parameters evaluated indicated that the developed D-μSPE protocol has high accuracy, satisfactory reproducibility, and practicality for the simultaneous analysis of 21 pesticides in tea beverages.

**Table 2.** The method performance of 21 targets in D-μSPE process.

| No. | Compound Name | 0.1 ng·mL⁻¹ | | 0.5 ng·mL⁻¹ | | 2.5 ng·mL⁻¹ | | LOD (ng·mL⁻¹) | LOQ (ng·mL⁻¹) |
|---|---|---|---|---|---|---|---|---|---|
| | | Average Rec/% | RSD /% | Average Rec/% | RSD /% | Average Rec/% | RSD /% | | |
| 1 | bupirimate | 62.1 | 4.1 | 64.4 | 2.5 | 63.6 | 2.6 | 0.05 | 0.1 |
| 2 | butachlor | 80.0 | 6.8 | 78.7 | 4.4 | 72.0 | 6.2 | 0.05 | 0.1 |
| 3 | cadusafos | 84.0 | 6.1 | 91.0 | 4.6 | 74.2 | 4.7 | 0.05 | 0.1 |
| 4 | coumaphos | 78.4 | 4.5 | 78.3 | 3.8 | 76.0 | 3.9 | 0.05 | 0.1 |
| 5 | demeton | 87.7 | 8.0 | 89.4 | 4.8 | 87.2 | 3.1 | 0.05 | 0.1 |
| 6 | disulfoton | 81.8 | 10.8 | 80.1 | 9.1 | 79.4 | 12.6 | 0.05 | 0.1 |
| 7 | disulfoton sulfone | 96.8 | 4.1 | 96.3 | 2.8 | 88.7 | 2.1 | 0.05 | 0.1 |
| 8 | disulfoton sulfoxide | 78.6 | 3.3 | 81.3 | 1.4 | 74.7 | 2.7 | 0.05 | 0.1 |
| 9 | ethoprophos | 71.0 | 3.5 | 74.6 | 3.5 | 72.4 | 3.3 | 0.05 | 0.1 |
| 10 | fensulfothion | 84.8 | 4.3 | 89.1 | 2.6 | 85.5 | 3.2 | 0.05 | 0.1 |
| 11 | fensulfothion oxon sulfone | 91.6 | 4.1 | 91.0 | 2.1 | 90.0 | 3.1 | 0.05 | 0.1 |
| 12 | fensulfothion sulfone | 89.9 | 4.1 | 90.7 | 3.7 | 86.6 | 2.6 | 0.05 | 0.1 |
| 13 | fluopyram | 75.3 | 3.6 | 78.4 | 3.0 | 71.6 | 3.1 | 0.05 | 0.1 |
| 14 | myclobutanil | 86.2 | 7.0 | 89.6 | 1.7 | 84.7 | 2.5 | 0.05 | 0.1 |
| 15 | phorate sulfoxide | 80.8 | 2.3 | 84.0 | 2.0 | 78.6 | 2.1 | 0.05 | 0.1 |
| 16 | prochloraz | - | - | 62.3 | 6.2 | 62.7 | 4.9 | 0.10 | 0.2 |
| 17 | profenofos | 79.7 | 3.7 | 80.4 | 3.8 | 74.1 | 2.8 | 0.05 | 0.1 |
| 18 | propiconazole | 106.6 | 5.0 | 89.7 | 3.1 | 79.8 | 3.9 | 0.05 | 0.1 |
| 19 | pyridaphenthion | 79.3 | 2.2 | 80.2 | 1.7 | 75.7 | 3.1 | 0.05 | 0.1 |
| 20 | terbufos sulfoxide | 75.0 | 3.8 | 77.2 | 2.4 | 72.8 | 3.7 | 0.05 | 0.1 |
| 21 | tetraconazole | 71.7 | 5.4 | 73.1 | 3.4 | 69.7 | 3.3 | 0.05 | 0.1 |

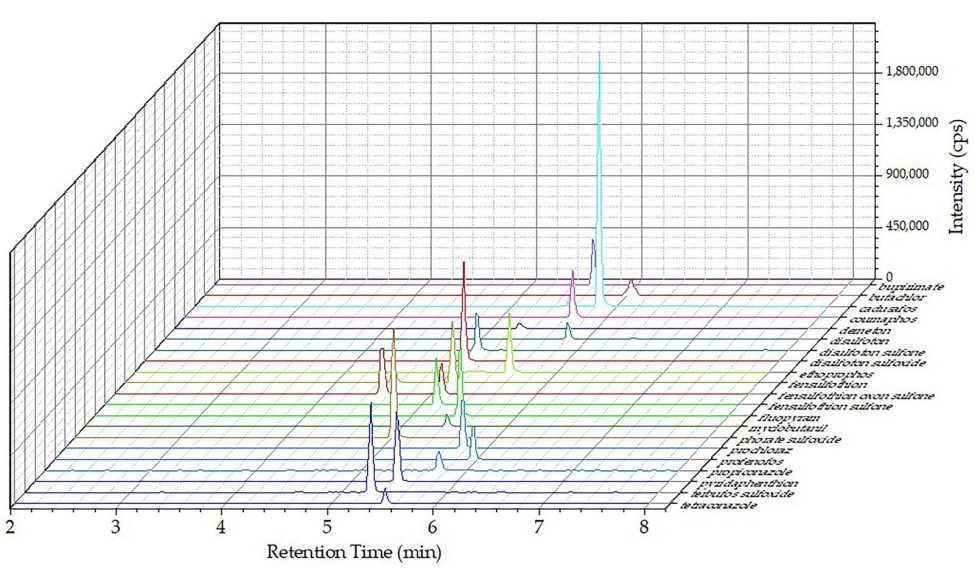

**Figure 5.** The chromatogram of 21 analytes at fortified level of 0.5 ng·mL⁻¹ in black tea beverage.

*3.7. Reusability of Zeolite H-ZSM-5*

Zeolite H-ZSM was reused to pre-concentrate 21 pesticides in black tea beverages (Figure 6) at a concentration of 2.5 ng·mL⁻¹ to assess reusability. Four replicates were carried out for each reuse. The results showed the mean recoveries for one to five times reuse in black tea at the range of 64.3–100.1%, 72.1–111.9%, 62.4–92.6%, 68.0–97.6%, and 66.1–97.6%, respectively. The RSDs of the five mean recoveries for each pesticide ranged from 2.1% to 12.7%. Twenty-one pesticides showed acceptable method accuracy and consistent deviations, demonstrating the feasibility of the regeneration procedure in general. Therefore, the regeneration of zeolite H-ZSM still has good methodological stability. In the previous study, H-beta zeolite was regenerated ten times to adsorb neonicotinoid pesticides from water, but was not applied to more complex honey samples [20]. The reusability

performance after washing the UiO-66 crystals with 3 mL of acetone showed that the extraction rate remained almost stable after ten cycles, except for thiamethoxam, which was below 60% [12].

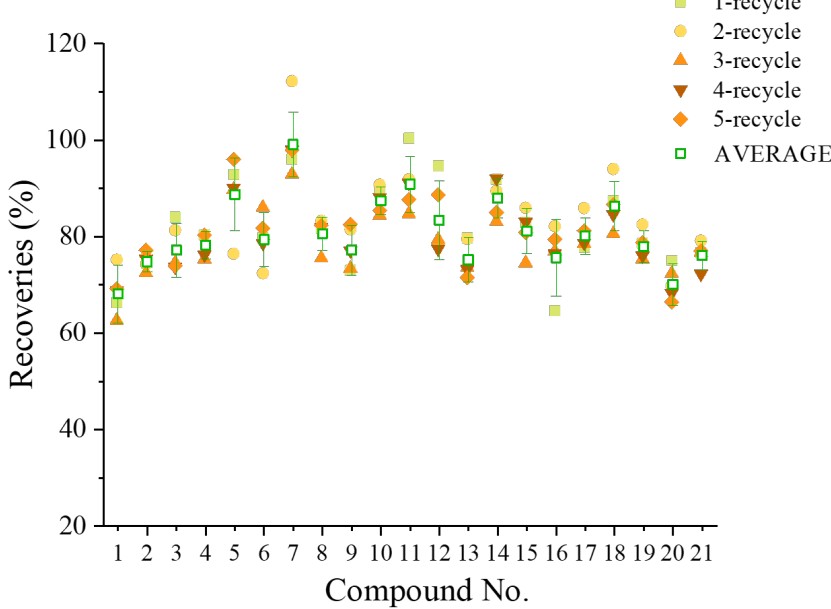

**Figure 6.** Recovery performance of five regenerations for 21 targets in black tea.

We found that H-ZSM-5 and H-Beta have similar porous crystal structure characteristics through characterization, providing favorable contact opportunities for the adsorption of OPs. Therefore, this study provides a convincing case for a gap in the zeolite adsorption of a wide range of OPs. In a series of optimizations of adsorption and desorption parameters, the ultra-rapid adsorption at 1 min showed a clear advantage in the studies of novel adsorbents that have been reported. It demonstrates that zeolite H-ZSM-5 has strong stability and compatibility in complex matrices. Additionally, due to the unique pore characteristics, it is possible to expel most of the organic matter disturbances in the matrix on the zeolite surface. The enhancement or weakening of the signal due to matrix effects during ionization of the sample can be effectively reduced. Therefore, the H-ZSM-5 can also be considered an option for purification procedures for complex matrices.

## 4. Conclusions

In summary, the prepared zeolite H-ZSM-5, with excellent crystallinity and abundant microporous characteristics, provided sufficient space and sites for the adsorption reaction. The H-ZSM-5, with a porous surface and a powdered state, has been used to adsorb pesticides directly from tea beverages. It showed fast and stable physical interactions of sixteen OPs and five other pesticides. Based on the advantages of rapid adsorption, the synthetic zeolite H-ZSM-5 was proposed as an environmentally friendly adsorbent for determining 21 pesticide residues in tea beverages. The D-µSPE pre-concentration procedure of Zeolite H-ZSM-5 as an adsorbent combined with UPLC-MS/MS showed acceptable analytical performance for 21 pesticides in tea beverages. The main advantages of our proposed Zeolite H-ZSM-5 based D-µSPE over many newly synthesized or modified adsorbents are the fast processing and the ability to extract and concentrate multiple targets simultaneously without problems of equilibration, time-consuming sample loading, and wash wastage. Although the internal mechanism of OPs adsorption by H-ZSM-5 zeolites is not fully explained, our proposed method provides an acceptable proposal for the rapid determination of trace pesticide residues in complex beverages. Further, the method significantly reduces the toxicity of organic solvents to the analyst and the environment and demonstrates analytical method reliability and stability in terms of reusability performance.

Alternatively, covering a wider range of OPs or pesticides in D-µSPE and other complex matrices for pesticide analysis can be further explored in the future.

**Supplementary Materials:** The following supporting information can be downloaded at: https://www.mdpi.com/article/10.3390/pr11041027/s1, Figure S1: The percentage of adsorption for time evolution; Figure S2: The matrix effects of 21 pesticides in black tea (BT) and green tea (GT) beverages; Table S1: Chemical information and Multiple Reaction Monitoring parameters for 21 pesticides; Table S2: The linearity and calibration curves of twenty-one pesticides.

**Author Contributions:** Conceptualization, W.S. and S.W.; methodology, B.B., C.Z., N.W. and S.W; software, N.W. and S.W; validation, N.W., X.J. and L.C.; formal analysis, B.B.; investigation, H.L.; resources, W.S.; data curation, H.Y. and Z.H.; writing—original draft preparation, B.B. and S.W.; writing—review and editing, W.S. and S.W.; visualization, N.W.; supervision, W.S.; funding acquisition, W.S. and N.W. All authors have read and agreed to the published version of the manuscript.

**Funding:** This research was funded by Chuying Project from Institute for Agri-food Standards and Testing Technology, Shanghai Academy of Agricultural Sciences (Grant No. Chuying 2022-1-4), and The APC was supported by Regional Public Brand of Chongming Agricultural Products: Construction of a Whole Industry Chain Standard System funded by Shanghai Agricultural and Rural Committee in Chongming District (Grant No. 2021CNKC-05-04) and Research on Integrated Prevention and Control Technology and Ecosystem Construction Demonstration of Paddy Field Surface Source Pollution funded by Shanghai Agricultural Science and Technology Promotion (Grant No. (2022) No. 2-3).

**Data Availability Statement:** The original contributions presented in the study are included in the article/Supplementary Material, further inquiries can be directed to the corresponding author.

**Acknowledgments:** We sincerely appreciate the language help and proofreading provided by Kaixuan Huo. All individuals included in this section have consented to the acknowledgement.

**Conflicts of Interest:** The authors declare no conflict of interest.

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
