# Peer review of "Development of a Zeolite H-ZSM-5-Based D-μSPE Method for the Determination of Organophosphorus Pesticides in Tea Beverages"

_processes, doi:10.3390/pr11041027_

Round 1
Reviewer 1 Report
Authors described a novel dispersive micro-solid phase extraction technique with H-ZSM-5 zeolite as an adsorbent for the determination of pesticides ina tea beverages.
The use of D-uSPE is interesting because it significantly reduces the consumption of toxic organic reagents. However, the production of the sorbent itself is complicated. I believe that the presented method is innovative and the article should be published in Processess. However, I have a few minor comments:
- authors should check and correct minor spelling and punctuation errors in the text.
- Figures 2 and 4 - the authors use the letters A and B in the figures, while in the description under the figures there is no indication which is figure A and which is B.
-The authors describe the developed chromatographic method used for the analysis of pesticides and provide recoveries for analytes and RSD%. Perhaps it would be worth presenting a chromatogram for all analytes (e.g. a sample fortified with 21 pesticides) and the real sample in which the D-uSPE technique was able to detect the presence of pesticides.
Reviewer 2 Report
Dear authors,
The manuscript entitled “Development of a zeolite H-ZSM-5-based D-μSPE method for the determination of organophosphorus pesticides in tea beverages” report on a novel dispersive micro-solid phase extraction (D-μSPE) technique with 15 H-ZSM-5 zeolite as an adsorbent was developed for the determination of 21 trace pesticides in tea beverages. It presents scientific relevance for the area of Chemistry, Biotechnology, Medicine and others area. After consulting www.sciencedirect.com; https://pubmed.ncbi.nlm.nih.gov/ and others data bases, authors have publications related to subjects related to the theme of the manuscript. The language (English) are satisfactory (I suggest the final revision)! However, you need to change some details/information in the abstract, Introduction, Methods, results, discussion and conclusions.
Abstract:
- The abstract is well written! Details of the methods (optimal parameters) used must be provided. I suggest inserting more numerical data about the results obtained!
- I suggest inserting, at the end, the advantages of the methods applied and the innovation of study for science.
1. Introduction section: It is well written, but I suggest highlighting the "innovative" proposal of the study, as well as the advantages / disadvantages, at the end of the introduction.
2. Material and Methods section: The methodological proposal is appropriate to the manuscript, but I suggest:
- Page 2, lines 90 and 92, in “2.1 Reagents and Materials” section: To replace “mg/L” and “ug/mL” by “mg L-1” and “ug mL-1”, and throughout the manuscript.
- Page 3, in “2.2 Preparation of Zeolite H-ZSM Adsorbent” and “2.3 Characterization of Zeolite H-ZSM” sections: How did you define these parameters? I suggest indicating the references used for the procedures adopted.
- Page 3, line 117, in “2.3 Characterization of Zeolite H-ZSM” section: "ICP-AES" or "ICP AES"? Or "ICP OES"?
- Page 3, lines 126 and 127, in “2.4 HPLC-MS/MS Analysis” section: To replace “mmol/L” and “mL/min” by “mmoL L-1” and “mL min-1”, and throughout the manuscript. How were these parameters (mobile phases, flow rate, volume fraction, etc…) optimized? Multivariate analysis?
- Page 3, lines 141 and 143, in “2.5 DSPE process” section: To replace “g/min” by “g min-1”, and throughout the manuscript. I suggest indicating the references used for the procedures adopted.
- Page 4, in “2.6 Sample Collection” section: How were the samples stored until the time of analysis? What is the storage time?
- Which concentration ranges were studied? What are the analytical validation parameters used? Has the proposed method been validated? If so, which protocol / guidelines did you follow? What are the validation parameters studied? Precision, accuracy, LOD, LOQ, robustness, etc. What concentration levels are used to assess accuracy? I suggest detailing the proposed method in more detail...
3. Results and discussion
I suggest expanding the discussions - better describe the findings and compare them with other works published in the literature!
- Page 8, in “3.6 The Validation of D-μSPE process” section: Some information in this section can be allocated in the "materials and methods"!
- Page 8, lines 289 and 297, in “3.6 The Validation of D-μSPE process” section: To replace “ng/mL and μg/L” by “ng mL-1 and μg L-1”, and throughout the manuscript.
- Page 8, in Table 2, in “3.6 The Validation of D-μSPE process” section: Are LOD and LOQ values ​​the same for all analytes? How were they calculated? I suggest review!
I suggest expanding the discussions based on the results obtained and the literature. The authors obtained interesting results that need to be further discussed.
- I suggest, at the end of the section, to write a paragraph summarizing the findings and their impacts on the research proposal.
- Conclusion section: I suggest highlighting the "innovative" proposal of the study, as well as the advantages/ disadvantages/limitations.
* Figures and Tables: Adequate! See remarks on Table 2!
* Supplementary Figures and Tables: Adequate!
* References: Please, check if the references are in accordance with the journal's rules.
Round 2
Reviewer 2 Report
The authors made the suggested corrections.